# Biological Activity of Different Forms of Oxidized Parathyroid Hormone

**DOI:** 10.3390/ijms232012228

**Published:** 2022-10-13

**Authors:** Ahmed A. Hasan, Carl-Friedrich Hocher, Burkhard Kleuser, Bernhard K. Krämer, Berthold Hocher

**Affiliations:** 1Institute of Pharmacy, Freie Universität Berlin, 14195 Berlin, Germany; 2Fifth Department of Medicine (Nephrology/Endocrinology/Rheumatology/Pneumology), University Medical Centre Mannheim, University of Heidelberg, 68167 Mannheim, Germany; 3Klinik für Innere Medizin, Bundeswehrkrankenhaus Berlin, 10115 Berlin, Germany; 4European Center for Angioscience, Medical Faculty Mannheim, University of Heidelberg, 68167 Mannheim, Germany; 5Key Laboratory of Study and Discovery of Small Targeted Molecules of Hunan Province, School of Medicine, Hunan Normal University, Changsha 410081, China; 6Reproductive and Genetic Hospital of CITIC-Xiangya, Changsha 410008, China; 7Institute of Medical Diagnostics, IMD Berlin, 12247 Berlin, Germany

**Keywords:** parathyroid hormone, chronic kidney disease-mineral bone disorder (CKD-MBD), oxidative stress

## Abstract

Preclinical studies have shown that parathyroid hormone (PTH) loses its biological effects through oxidation. PTH can be oxidized at methionines 8 and 18. Three possible variations of oxidized PTH (oxPTH) exist: Met8(ox)PTH, Met18(ox)PTH, and Met8, Met18(di-ox)PTH. A recent study showed that Met18(ox)PTH retained biological activity and was able to upregulate *Fgf23* gene expression, whereas Met8(ox)PTH and Met8, Met18(di-ox)PTH showed less or no biological activity. An earlier study likewise showed that the oxidation of Met18 has minor effects on the secondary structure of PTH, whereas the oxidation of Met8 causes substantial structural changes, consistent with another study showing that oxidization just at Met8 blocks the generation of the second messenger cAMP, whereas the effect of the oxidation of Met18 is much less potent in inhibiting cAMP formation. A considerable percentage of circulating PTH in chronic kidney disease (CKD) patients is oxidized. However, we do not know the relative amounts of the different forms of oxPTH with agonistic, partial agonistic, or even antagonistic biological actions in different CKD populations. This might explain different clinical findings in the different CKD populations analyzed so far. The currently available method that was used in these clinical studies just distinguishes between oxPTH and noxPTH without being able to differentiate between different forms of oxPTH. Only methods of PTH measurement that are able to differentiate between PTH forms (noxPTH, Met8(ox)PTH, Met18(ox)PTH, and Met8, Met18(di-ox)PTH) have the potential to improve patient care, because only these methods will definitively separate bioactive from non-bioactive PTH forms. Such methods need to be developed, validated, and used in prospective randomized clinical trials to define the potential value of bioactive PTH forms as a predictor of cardiovascular events, mortality, and bone turnover.

## 1. Introduction

Parathyroid hormone (PTH) is a key hormone involved in regulating calcium and phosphate homeostasis. The main trigger for the parathyroid gland to produce and release PTH is a low level of non-protein-bound calcium in the blood. In order to correct this hypocalcemia, PTH exerts many biological actions on different target organs, mainly through activating its receptor, PTH1R. It stimulates the renal reabsorption of calcium, inhibits the renal reabsorption of phosphate, and activates renal 1α-hydroxylase to produce calcitriol (1,25 (OH)2D3 (the active form of vitamin D) [1,2]. Another important effect of PTH is its catabolic effect on bones by mobilizing calcium and phosphate into the circulation, leading to enhanced bone resorption. This bone catabolic effect is seen when PTH blood levels are stably elevated over a long time, for instance, in chronic kidney disease patients. This syndrome is often referred to as chronic kidney disease–mineral bone disorder (CKD-BMD). However, the intermittent administration of PTH is believed to exert a bone anabolic effect, which is why the PTH analog teriparatide is approved for the treatment of osteoporosis [3]. Figure 1 summarizes the roles of the key players and organs in CKD-BMD. One of these key players is fibroblast growth factor 23 (FGF23), which is produced by osteocytes, and its main biological effect is phosphaturia. PTH can upregulate FGF23 expression, and FGF23 was reported to inhibit PTH production in a negative feedback loop [4]. One of the main disorders of calcium and phosphate dysregulation is ectopic calcification, which is the deposition and formation of insoluble calcium phosphate salts in soft tissue, leading to malfunction. The calcification of cardiovascular tissues, such as the aorta, is thought to be the hidden culprit for increased cardiovascular mortality in patients with CKD-MBD [5]. In summary, PTH is involved in many clinical situations, such as osteoporosis, CKD-BMD, and ectopic calcification; thus, the precise detection and quantification of PTH in the blood would be of great value in this regard.

## 2. Chronic Kidney Disease–Mineral Bone Disorder (CKD-MBD) and Uremic Calcification

CKD-MBD is a systemic disorder of mineral and bone metabolism that develops as a consequence of CKD. It is characterized by a combination of biochemical irregularities, bone metabolism disturbances, and vascular calcification. Disturbed mineral metabolism is often associated with CKD and is a major contributing factor during the remodeling of the vascular system and bone. CKD-MBD may include disorders of calcium, phosphorus, PTH, and/or vitamin D metabolism, reduced GFR, vascular calcification, and disturbed bone growth [6,7]. Glomerular filtration is normally followed by the reabsorption of phosphorus in the proximal tubule. Normal phosphate levels are maintained through the action of Klotho (a transmembrane protein produced mainly in the kidney), PTH, and FGF23. In the early stages of CKD, however, the kidney is not able to excrete appropriate amounts of phosphate. FGF23 and PTH reduce the expression of sodium/phosphate cotransporters in the proximal renal tubules. This prevents phosphate reabsorption from the filtrate to the bloodstream. The normal functioning of FGF23, a protein secreted from osteoblasts and osteocytes, depends upon the presence of the co-factor Klotho. Vitamin D is transformed into calcitriol, the active form of vitamin D, in the renal tubules by 1α-hydroxylase. FGF23 lowers the levels of calcitriol by inhibiting the activity of 1α-hydroxylase, which converts 25(OH)-vitamin D3 (calcidiol) into its active form, 1,25(OH)2-vitamin D3 (calcitriol). Furthermore, FGF23 stimulates 24-hydroxylase, which deactivates calcitriol by further hydroxylating it into calcitroic acid. Vitamin D improves serum calcium levels by boosting intestinal calcium absorption, reducing renal calcium discharge, and elevating calcium resorption from the bone [8,9]. PTH also helps raise serum calcium levels by aiding calcitriol formation in the kidneys and increasing the tubular reabsorption of calcium from the glomerular filtrate, thus preventing urinary calcium loss. The occurrence of CKD also results in early calcitriol insufficiency and high levels of FGF23. Low calcitriol contributes to hypocalcemia and, together with the established hyperphosphatemia, causes increased PTH release and secondary hyperparathyroidism. As CKD worsens, the concentration of phosphate in the exchangeable pools rises, resulting in an unfavorable phosphate equilibrium, which, in turn, stimulates the production of FGF23 and PTH secretion as a homeostatic reaction. The routine function of the skeleton as a phosphate reservoir when phosphate equilibrium is positive is observed in various syndromes of hyperphosphatemia in mammalian pathophysiology [10,11]. In CKD, excessive bone resorption rates versus bone formation rates exacerbate the condition. Accordingly, the skeleton contributes to hyperphosphatemia in CKD, and the reservoir function of the skeleton that should act in the presence of a positive phosphorus balance is halted. Chronically raised levels of PTH initiate severe bone resorption, further intensifying hyperphosphatemia, as the normal ‘storage container’ feature of the skeleton is lost. The vasculature, along with the soft tissues, functions as an alternative reservoir for excess phosphate, which may eventually result in vascular calcification [7,8].

One of the most common causes of death in patients with CKD is CVD. Uremic vascular calcification might be the primary reason for poor CVD outcomes in patients with CKD [12,13]. Vascular calcification was thought to be a passive and degenerative condition without any proposed efficient treatment solutions. The exact mechanisms by which vascular calcification progresses are yet to be fully mapped. A better understanding of the molecular pathways responsible for this pathological mineralization would allow for the creation of novel drug targets that might help to reduce the harmful effects of vascular calcification in patients with CKD. Vascular calcification might be an active, tightly controlled cell-mediated process that bears a resemblance to bone mineralization [5]. Phosphate is a critical molecule in this process. Phosphate is principally delivered to the smooth muscle cells by the ubiquitous type III sodium phosphate transporters. Elevated intracellular phosphate typically leads to the increased expression of osteochondrocyte transcription factors, such as RUNX2, Osterix, MSX2, and SOX9, provoking a phenotypic transition in vascular smooth muscle cells (VSMCs) from a contractile phenotype, identified by markers of the smooth muscle lineage (SMAD6, matrix Gla protein, smooth muscle actin, αSMA, and SM22α), to an osteoblastic-like phenotype, identified by osteopontin, bone sialoprotein II, osteonectin, collagen I/II and osteocalcin, and mineralization-competent matrix vesicles. The transdifferentiation of VSMCs into osteoblast-like cells results in mineral deposition in the ECM, the loss of contractile properties, and apoptosis. This phenotypic alteration results in the increased deposition of calcium and phosphate crystals, mimicking what occurs during bone formation [5,14].

In CKD, VSMC injury is a result of irregular mineral homeostasis, in which elevations in calcium and phosphate have a synergistic impact. After VSMCs are injured, apoptosis and vesicle formation occur as a result. This provokes the phenotypic transformation of VSMCs into osteoblast-like cells, as mentioned before. Apoptotic bodies also serve as a nidus for calcification in the vessel wall. Calcification is accelerated in the presence of high levels of PTH, vitamin D, calcium, and, most significantly, phosphate. While vascular calcification is common in CKD and ESRD, not all patients experience calcification, despite persistent abnormalities in mineral homeostasis. This has led to the identification of several local and systemic inhibitors of vascular calcification that protect against calcification in healthy individuals and certain patients with CKD [11,15].

A variety of potential treatment possibilities have been reported for vascular calcification. Nevertheless, the similarity between vascular calcification and bone development makes it more complicated to specifically target vascular calcification without posing adverse effects on bones and teeth. CKD patients should restrict their dietary phosphate consumption while sustaining an adequate protein intake. The form of phosphate is also essential. Inorganic phosphate present in food additives, e.g., in fast food, is more efficiently absorbed compared to organic phosphate. Hence, CKD patients are also asked to manage the quality of their diet. Furthermore, inorganic phosphate is cleared during dialysis, and therefore, increasing the dialysis session frequency and length could help to reduce hyperphosphatemia. Other dialysis factors can also alter the clearance of inorganic phosphate, such as the blood and dialysate flow rate and dialyzer membrane surface area. Moreover, to effectively lower serum inorganic phosphate levels, a variety of phosphate binders are used in CKD patients. Phosphate-binding agents are typically recommended for the treatment of individuals with ESRD to restrict the intestinal absorption of this ion. The currently offered phosphate binders involve calcium or magnesium salts. They all lower inorganic phosphate serum levels efficiently, although calcium-free phosphate binders were shown to be more efficient in delaying vascular calcification progression. Calcimimetic therapies, such as cinacalcet, mimic the action of calcium on tissues by allosteric activation of the calcium-sensing receptor. These therapies decrease serum PTH as well as calcium and phosphate levels. These agents appear to decrease arterial calcification related to CKD. Bisphosphonates (pyrophosphate analogs), utilized as a typical treatment for osteoporosis, have been regarded as a possible vascular calcification therapy choice because of their inhibitory effect on hydroxyapatite (a calcium and phosphate complex) crystal formation [5,10,13,15].

## 3. Oxidative Stress in CKD

Despite the great improvement in dialysis technologies, the life expectancy of patients with end-stage renal disease (ESRD) undergoing chronic hemodialysis (HD) is still rather poor when compared to the general population [12,13,16]. In patients with chronic renal failure, cardiovascular complications increase as renal function deteriorates, and the risk of reaching ESRD is as present as the risk for cardiovascular disease. Once ESRD is reached, cardiovascular disease (CVD) is the leading cause of morbidity and mortality, and about half of the deaths of patients on dialysis are attributed to cardiovascular causes [12,13,16]. Several contributors to the complex puzzle of uremia-related cardiovascular disease have been identified so far, with oxidative stress being one of the most important players [17].

The etiology of oxidative stress is multifactorial, and frequent conditions in ESRD contribute to the increased pro-oxidant activity, including age, diabetes mellitus, malnutrition, uremia, and chronic inflammation. Several interrelated processes occur in renal insufficiency, which may lead to the accelerated development of atherosclerosis in uremia, such as oxidative stress, inflammation, endothelial dysfunction, and vascular calcification. Beyond this, the antioxidant system is severely impaired in uremic patients and gradually altered with the degree of renal failure. A more detailed understanding of the connection between oxidative stress and other risk factors is of therapeutic importance and could help to identify new markers of an increased risk of cardiovascular complications in patients on dialysis [18,19].

Metabolic activity in biological systems such as the human body requires energy-transducing processes. Oxidation–reduction reactions occur when molecules undergo chemical changes during the synthesis and degradation of carbohydrates, proteins, and lipids. The dysregulation of metabolic reactions may occur in pathological states, which could cause an imbalance between the production of reactive oxygen species (ROS) and the local antioxidant capacity, a status known as oxidative stress. Several control and defense mechanisms exist to protect the body from the damaging effects of ROS. Deficiencies in the antioxidant defense system can play a crucial role in the pathogenesis of different renal diseases. Antioxidative enzyme systems, such as superoxide dismutase, glutathione peroxidase, and catalase, inactivate free radicals under physiological conditions, thus preventing tissue damage. ROS are partially reduced and potentially toxic reactive oxygen intermediates, which have an unpaired electron in their outer orbital. Along with reactive nitrogen species (RNS), they have a complex role in health and disease and are thought to contribute to the development of a wide range of diseases, including CVDs. Under physiological conditions, ROS/RNS play an important role as part of the defense system against invading microorganisms and malignant cells. They are also important in signaling pathways that regulate vascular tone and in cellular processes such as proliferation, migration, and differentiation. However, as a response to metabolic dysregulation, the excessive activation of oxidative processes may lead to oxidative stress. The resulting oxidation of important macromolecules, including proteins, lipids, carbohydrates, and DNA, ultimately leads to tissue injury [20,21].

Patients on dialysis usually show a correlation between markers of inflammation and markers of oxidative stress, as well as a negative correlation between inflammation and antioxidative defense mechanisms. In fact, patients with ESRD live in a chronic inflammatory state, which derives from the loss of kidney function on the one hand and from dialysis treatment on the other hand. With the impairment of kidney function, several substances accumulate in the plasma. The increased production of inflammatory mediators in uremia and the retention of cytokines, pro-oxidants, and uremic toxins add to a chronic state of inflammation. Treatment with dialysis puts a number of additional risk factors on patients, such as the bio-incompatibility of membranes and dialysate impurity, as well as the removal of antioxidants that occurs during the procedure [16,22].

It can be assumed that the interplay between inflammation and oxidative stress promotes the deterioration of renal function as well as the increased development of CVD observed in ESRD. However, it is not clear whether, during uremia, chronic inflammation causes the elevation of oxidative stress or if oxidative stress is responsible for activating the immune system through a variety of inflammatory stimuli related to uremia and dialysis. Oxidative stress is usually higher in hemodialysis patients with cardiovascular disease [17,23].

In the pathogenesis and progression of cardiovascular disease, several oxidative modifications contribute to different manifestations of CVD. Oxidative stress may play an important role in the initiation and development of atherosclerosis, as it induces endothelial dysfunction by impairing the bioactivity of endothelial nitric oxide and promotes leukocyte adhesion, inflammation, thrombosis, and smooth muscle cell proliferation. Since oxidized LDL plays a relevant role in the overall process of atherosclerosis, the increased oxidation of lipids could represent a critical step in the development of cardiovascular diseases in the ESRD population. In addition to the direct adverse effects on the kidney, oxidized LDL mediates several proinflammatory and proatherogenic processes, such as the proliferation of smooth muscle cells and the differentiation of monocytes in the arterial wall. It accumulates in macrophages, which leads to foam cell generation in vascular walls and ultimately ends in the formation of atherosclerotic plaques. The oxidative modification of LDL is considered a major contributor to the development of atherosclerosis, and oxidized LDL can be found in human atherosclerotic lesions. Healthy subjects with the highest levels of oxidized LDLs had a significantly higher cumulative risk of cardiovascular disease. In uremic patients, this relation might even be more pronounced because of the increased susceptibility of LDL to oxidation [17,23].

DNA is very susceptible to oxidative stress, and patients with CKD show increased levels of genetic damage, mainly due to oxidative stress. Oxidative genetic damage increases when renal function decreases and is the highest in hemodialysis patients [24]. The oxidation of DNA can lead to mutations and genomic instability. High levels of genetic damage in CKD might be associated with a high incidence of cardiovascular pathologies, such as atherosclerosis and endothelial dysfunction [25].

## 4. Oxidation of PTH and Its Available Assays

PTH mRNA is translated into pre-pro-PTH, composed of 115 amino acids, in the parathyroid gland. The amino terminus of pre-pro-PTH is then cleaved to produce the 90-amino-acid pro-PTH, which is then further processed into the active mature PTH (1-84) [26]. The synthetic PTH fragment (1-34) retains total activity; thus, it is widely used in research due to technical and cost aspects. Human PTH contains two methionines at position 8 and position 18. These two methionines are prone to oxidation, leading to the formation of three different forms of oxidized PTH (oxPTH): Met8(ox)PTH, Met18(ox)PTH, and Met8, Met18(di-ox)PTH. Figure 2 shows the amino acid sequences of PTH (1-84) across different species. It can be noted that both Met8 and Met18 are conserved in human, mouse, and bovine PTH. However, rat and porcine PTHs lack Met18. This lack might support the notion, which is discussed later, that Met18 is less important for PTH biological activities than Met8. Figure 3 presents the binding sites for the available assays for PTH detection. First-generation assays used only one antibody directed at epitopes in the C-terminus of PTH. These assays measured a mixture of active PTH and inactive C-terminal fragments. Then, second- and third-generation assays were developed. These assays are sandwich enzyme-linked immunosorbent assays, usually utilizing two antibodies, one directed to the N-terminus and the second intended to bind the C-terminus of PTH. These assays are also known as intact PTH (iPTH) assays, specifically detecting the whole-length PTH. Second- and third-generation assays are routinely used in clinical settings, even though they cannot distinguish between non-oxidized (noxPTH) and oxPTH [27]. We developed a method to measure noxPTH by removing oxPTH using affinity chromatography before applying a third-generation assay [28]. Briefly, plasma samples are applied on a MobiSpin column containing immobilized monoclonal antibody raised against human oxidized PTH (1-34) fragments. The flow-through, wash fractions, and eluate of the column are collected and lyophilized. Then, the samples are reconstituted in suitable buffers and analyzed by the Roche Elecsys^®^ PTH, Intact (Roche, Penzberg, Germany) assay [28].

There is broad debate on whether measuring noxPTH would add important information for making clinical decisions. It should also be noted that our method of measuring noxPTH removes all oxPTH and cannot distinguish between Met8(ox)PTH, Met18(ox)PTH, and Met8, Met18(di-ox)PTH, which might have residual or different biological activity. After its release from the parathyroid gland, parathyroid hormone is metabolized in the liver into several fragments. Taking into account that (1-34) PTH is believed to be the biologically active fragment and thus used in in vitro experiments for logistic reasons, full-length (1-84) PTH and amino-terminal PTH fragments account for 18% of circulating PTH [PMID: 30627584]. The second-generation PTH assay recognizes full-length (1-84) PTH and the (7-84) C-PTH fragment, while the third generation recognizes only full-length (1-84) PTH. The biological activity of oxidized and non-oxidized PTH fragments still needs further investigation, taking into account that these fragments accumulate in CKD patients, and some of them were even reported to have antagonistic activity [PMID: 30627584 and PMID: 32178977]. Only high-resolution mass spectrometric methods can clarify this open research question. We also assume that CKD patients suffer from more oxidative stress in comparison to non-CKD patients. Thus, CKD patients might have not only a high total concentration of PTH but also a high percentage of oxPTH compared to non-CKD patients. This might make measuring PTH and its standardization more difficult in CKD patients due to the high variability of the oxPTH percentage from patient to patient and in non-CKD patients due to oxPTH levels that are below the lower detection limits of conventional analytical methods.

In the following two sections, we review the literature on the biological activity of the different forms of oxPTH and the potential added clinical value of measuring noxPTH.

## 5. Preclinical Studies Investigating the Oxidation of PTH

Table 1 summarizes the preclinical studies that investigated the biological activities of the different forms of oxPTH. In silico modeling showed that oxidizing PTH might lead to 3D structural changes that might hinder effective interactions between PTH and its receptor [29]. In another study, the effect of PTH oxidation on its secondary structure was investigated using circular dichroism, a spectroscopic absorption method. Interestingly, the impact was seen in the following order: Met8, Met18(di-ox)PTH > Met8(ox)PTH > Met18(ox)PTH [30]. The biological activity of the oxidized forms of PTH was also investigated in many in vitro studies. One of the most commonly used readouts for this purpose was the ability of PTH to stimulate renal and bone adenylyl cyclase to produce the second messenger cAMP [31,32,33,34,35]. Other in vitro readouts also included the activity of alkaline phosphatase [36], vasodilatory action [37], the relaxation of trachea preparations [38], chronotropic as well as inotropic effects [39], the binding affinity as well as the activation of mitochondrial ATPase [40], and *Fgf23* gene expression [41]. These studies presented clear evidence that PTH oxidation led mostly to the loss of biological activity, and some of them [31,33,40,41] even showed that the position of oxidation mattered, where oxidizing Met8 had a higher impact. In vivo studies using parathyroidectomized rats [42,43], immature Japanese quail birds [44], and rabbits [34] showed comparable results, where oxPTH failed to induce PTH actions such as increased blood calcium, decreased blood phosphate, decreased urine calcium, increased urine phosphate as well as cAMP, increased renal 1,25 (OH)_2_D3, and decreased renal 24,25(OH)_2_D3.

## 6. Clinical Studies Investigating the Oxidation of PTH

About ten years ago, a method was developed to separate noxPTH from all oxidized forms of PTH. This method was validated by high-resolution nano-liquid chromatography coupled to ESI-FTT tandem mass spectrometry (nanoLC-ESI-FT-MS/MS) [28]. So far, a few observational clinical studies analyzing the association between noxPTH and outcomes have been reported (summarized in Table 2). These studies investigated different CKD populations, such as adults with stage 2–4 CKD, patients on hemodialysis with and without hyperparathyroidism, and patients after kidney transplantation. The largest of these studies has only been published as an abstract so far. Baseline samples from 2867 participants of the EVOLVE trial (ClinicalTrials.gov: NCT00345839, accessed on 15 August 2022) were analyzed. The patients were followed for up to 64 months. The primary composite endpoint was the time until death, myocardial infarction, hospitalization for unstable angina, heart failure, or a peripheral vascular event. Pearson’s correlation analyses showed a very strong relationship between iPTH and oxPTH (r = 0.996; *p* < 0.001) and a weaker relationship between iPTH and noxPTH (r = 0.82; *p* < 0.001). A multivariate Cox regression model adjusted for patient characteristics, cardiovascular comorbidities, and baseline clinical laboratory parameters revealed that noxPTH, but not oxPTH or iPTH, was associated with the EVOLVE primary composite endpoint (time until death, myocardial infarction, hospitalization for unstable angina, heart failure, or a peripheral vascular event; hazard ratio 1.078; 95% CI 1.012–1.148; *p* = 0.020), cardiovascular mortality (hazard ratio 1.111; 95% CI 1.014–1.218; *p* = 0.024), and all-cause mortality (hazard ratio 1.113; 95% CI 1.038–1.193; *p* = 0.003) [45]. In contrast, a very recent small study (*n* = 31) investigating the association between noxPTH, iPTH, and biochemical and morphological markers of bone turnover in CKD patients showed a similar association of iPTH and noxPTH with biochemical and morphological markers of bone turnover [46]. Furthermore, a study in CKD patients prior to dialysis reached similar conclusions. The degree of association of noxPTH and iPTH with cardiovascular endpoints was virtually identical [47]. On the other hand, studies in patients after kidney transplantation [48], as well as end-stage renal disease patients on dialysis [49], showed comparable findings to the analysis of the large EVOLVE study described above [45]. The measurement of noxPTH was superior to the measurement of iPTH regarding the association with cardiovascular and renal endpoints. Moreover, we recently reported that total PTH and oxPTH substantially increased, whereas noxPTH only moderately increased with the progressive deterioration of kidney function in two independent cohorts of children with CKD and stable renal transplant recipients [41]; this might mean that the increase in PTH with decreasing GFR is mainly due to an increase in oxPTH in more advanced stages of CKD.

## 7. Conclusions and Open Questions

Many clinical studies have shown that the associations between all-cause mortality, as well as other patient-level parameters (such as cardiovascular events, body weight, body mass index, and body fat), and iPTH are U-shaped (Figure 4), with increased risk at both ends, i.e., too low or too high iPTH [50,51,52,53,54]. Moreover, it was reported that iPTH shows an almost perfect correlation with oxPTH (Figure 4) and a relatively weaker correlation with noxPTH [41,45,48]. Thus, it seems that iPTH measurements reflect, to a large extent, oxidative stress as well, which is certainly also harmful to CKD patients and not just bioactive PTH. Based on this observation, we put forward the hypothesis that the iPTH mortality curve results from an overlap of two curves, an oxidative-stress-related mortality curve (hypothetically linear) and a bioactive-PTH-form-related mortality curve. Dissecting these two curves might help in optimizing treatment guidelines. In other words, measuring noxPTH (or biologically active PTH, if some oxPTH forms are still active) rather than iPTH might shift a large proportion of the studied populations to within the recommended target PTH levels. Unfortunately, the most commonly used methods measure only iPTH and do not take oxidized forms of PTH into account. Even the only currently available assay for measuring noxPTH excludes all of the oxidized forms of PTH, without considering their potential remaining or modified activity (agonistic or antagonistic). Therefore, a new assay is needed to differentiate between noxPTH and the different forms of oxPTH. Clinical studies are also needed to establish the association of the different forms of oxPTH with clinical outcomes. The findings of such clinical studies might help to explain the discrepancies among clinical studies that used the current noxPTH assay, which cannot differentiate between the different forms of oxPTH. Three possible variations of oxidized PTH (oxPTH) exist: Met8(ox)PTH, Met18(ox)PTH, and Met8, Met18(di-ox)PTH. It was reported that the biological activity might be in the following order: noxPTH > Met18(ox)PTH > Met8(ox)PTH > Met8, Met18(di-ox)PTH. The studies analyzing the association of noxPTH with clinical outcomes differ with regard to the analyzed populations: CKD patients on dialysis with hyperparathyroidism, CKD patients on dialysis without hyperparathyroidism, transplanted patients, pediatric CKD patients prior to dialysis, and CKD stage 2–4 patients. These different patient populations most likely have different relative ratios of concentrations of the individual PTH forms. In other words, the biological oxPTH activity in the different CKD populations could be different simply because the relative concentration of Met18(ox)PTH (partial agonist) is different [2,28]. PTH measurements that are able to differentiate between PTH forms (noxPTH, Met8(ox)PTH, Met18(ox)PTH, and Met8, Met18(di-ox)PTH) are the only methods that might improve patient care, because only these methods will definitively separate bioactive from non-bioactive PTH forms. Such methods need to be developed, validated, and used in prospective randomized clinical trials to define the potential value of bioactive PTH forms as a predictor of cardiovascular events, mortality, and bone turnover.

## Figures and Tables

**Figure 1 ijms-23-12228-f001:**
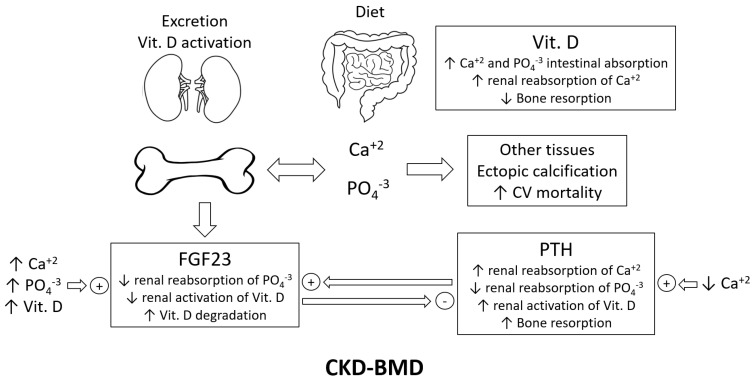
Key players in CKD-BMD; Vit. D = vitamin D; CV = cardiovascular; FGF23 = fibroblast growth factor 23; and PTH = parathyroid hormone.

**Figure 2 ijms-23-12228-f002:**
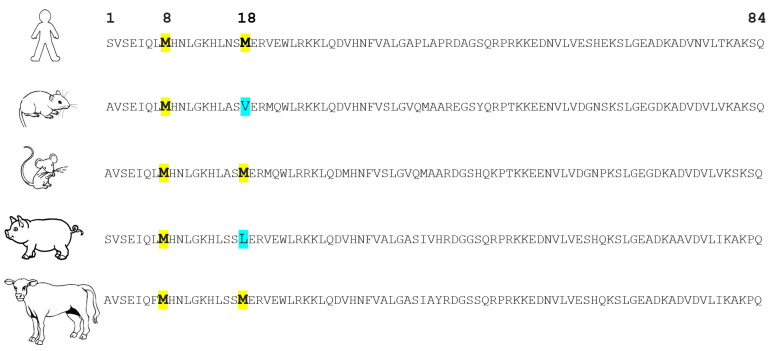
Amino acid sequences of parathyroid hormone (1-84) across different species. Amino acid sequences are adapted from https://www.uniprot.org/, accessed on 15 August 2022. The letters represent the IUPAC codes for amino acids. Amino acids at positions 8 and 18 (methionines in humans) are highlighted.

**Figure 3 ijms-23-12228-f003:**
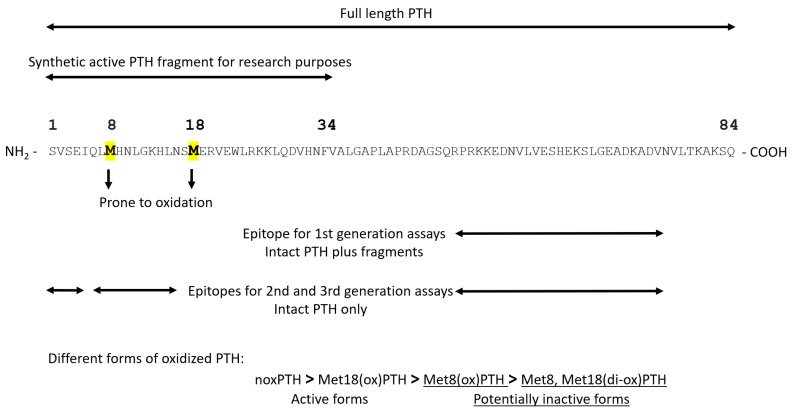
Structure of PTH and its available assays. The letters represent the IUPAC codes for amino acids; PTH = parathyroid hormone; noxPTH = non-oxidized PTH; Met = methionine; ox = oxidized; and di-ox = di-oxidized.

**Figure 4 ijms-23-12228-f004:**
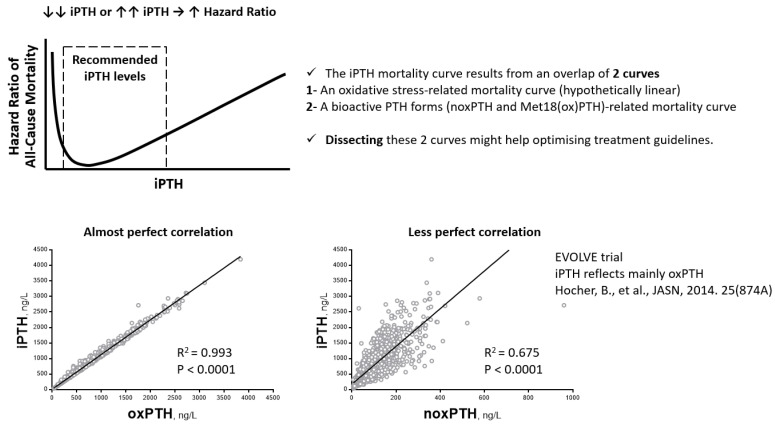
All-cause mortality curve of iPTH and its correlation with oxPTH and noxPTH. The U-shaped association is derived from the following publications: [50,51,52,53,54], and the correlation data are derived from the EVOLVE trial [45]. Other smaller studies [41,48] showed comparable correlations too. PTH = parathyroid hormone; oxPTH = oxidized PTH; noxPTH = non-oxidized PTH; Met18(ox)PTH: PTH oxidized at Met 18.

**Table 1 ijms-23-12228-t001:** Preclinical studies on oxPTH.

	Reference	Subject and/or Model	Readouts	Conclusions
1	[29]	In silico modeling	noxPTH and oxPTH interaction with PTH1R	PTH oxidation → 3D structure changes → altered PTH-PTH1R interaction
2	[30]	Circular dichroism	The secondary structure of PTH	Oxidation at Met18 → small impact Oxidation at Met8 → substantial changes Oxidation at both Met8 and Met18 → greater changes
3	[31]	In vitro renal membrane adenylyl cyclase assay	cAMP formation as a surrogate measure of renal adenyl cyclase activity	Oxidation at Met18 → full agonist with ↓ potency Oxidation at Met8 → partial agonist with ↓↓ potency Oxidation at both Met8 and Met18 → partial agonist with ↓↓↓ potency
4	[32]	In vitro plasma membranes purified from bovine renal cortex	Binding affinity and cAMP formation as a surrogate measure of renal adenyl cyclase activity	Oxidation of PTH → loss of the ability to bind plasma membranes and to activate adenylate cyclase
5	[33]	In vitro renal membrane adenylyl cyclase assay	cAMP formation as a surrogate measure of renal adenyl cyclase activity	The degree of bioactivity: noxPTH > Met18(ox)PTH > Met8(ox)PTH > Met8, Met18(di-ox)PTH
6	[34]	In vivo rabbits; In vitro isolated perfused rabbit proximal tubules	Renal electrolyte handling and adenylate cyclase stimulation	PTH → phosphaturia, anticalciuria, ↑ renal proximal tubular adenylate cyclase, and ↑ renal cortical cAMP Oxidized forms → weaker or no activity
7	[35]	In vitro adenylate cyclase assay using rat osteosarcoma cells	cAMP formation as a surrogate measure of adenyl cyclase activity	All oxidized forms possessed reduced biological activity, more so for oxidation at Met8 than at Met18.
8	[36]	In vitro primary cultures of neonatal mouse calvarial cells	The activity of alkaline phosphatase	Oxidation of PTH → loss of the ability to activate alkaline phosphatase
9	[37]	In vitro coronary, renal, hepatic, and visceral vascular beds	Direct vasodilatory action	noxPTH but not oxPTH (at Met8 and Met18) has direct vasodilatory action
10	[38]	In vitro guinea-pig trachea constricted with histamine in vitro	Relaxation of guinea-pig trachea constricted with histamine	noxPTH but not oxPTH (at Met8 and Met18) can relax guinea-pig trachea constricted with histamine
11	[39]	In vitro isolated frog atrium	Cardiac action (chronotropic and inotropic effects)	noxPTH → +ve chronotropic and inotropic effects oxPTH → abolished cardiac action
12	[40]	In vitro intact mitochondria, submitochondrial particles, and purified mitochondrial F1 ATPase	The affinity (specific binding) to mitochondrial ATPase and its activation	PTH → ↑ affinity and activation of the mitochondrial ATPase Oxidation of Met18 → 50% ↓ affinity Oxidation of Met8 → 95% ↓ affinity Oxidation of both Met8 and Met18 → further ↓ affinity
13	[41]	In vitro UMR106 osteoblast-like cells	*Fgf23* gene expression	noxPTH → ↑ *Fgf23* mRNA synthesis Oxidation of PTH, in particular, at Met8 → ↓ ↑ *Fgf23* mRNA synthesis
14	[42]	In vivo parathyroidectomized rats on low-calcium diet	Blood calcium	Oxidation of PTH → loss of the ability to increase blood calcium
15	[43]	In vivo parathyroidectomized rats and vit. D-deficient rats	Serum calcium, serum phosphate, urine calcium, urine phosphate, and urine cAMP, as well as renal 1,25 (OH)_2_D3 and 24,25(OH)_2_D3 production	noxPTH → ↑ serum calcium, ↓ serum phosphate, ↓ urine calcium, ↑ urine phosphate, ↑ urine cAMP, ↑ renal 1,25 (OH)_2_D3, and ↓ renal 24,25(OH)_2_D3 Oxidation at both Met8 and Met18 → loss of all above-mentioned biological activities
16	[44]	In vivo immature birds (Japanese quail)	Hypercalcemic response	Oxidation of PTH → no hypercalcemic response

**Table 2 ijms-23-12228-t002:** Clinical studies on oxPTH.

	Reference	Design	Individuals (n)	Population Characteristics	Conclusions	Remarks
1	[28]	Observational study	18	Patients on intermittent hemodialysis	The % of oxPTH was 7–34%	No follow-up for the patients; No investigation of the associations between the different forms of PTH and clinical outcomes
2	[45]	Retrospective study (EVOLVE trial, NCT00345839)	2867	Maintenance hemodialysis	Only noxPTH, but not iPTH or oxPTH, was associated with cardiovascular events and mortality	Follow-up for up to 64 months
3	[46]	Prospective observational bone biopsy study	31	Patients with ESKD and low bone turnover, normal bone turnover, or high bone turnover	Measuring noxPTH has no added value compared to total PTH as an indicator of bone turnover in patients with kidney failure	Bone turnover markers showed similar correlation coefficients to noxPTH and total PTH
4	[47]	Observational study	535	Patients with CKD	PTH was associated with all-cause mortality; there was no association of noxPTH with any of the clinical outcomes examined	Follow-up over 5.1 years for the occurrence of acute heart failure, atherosclerotic events, CKD progression, or all-cause death
5	[48]	Prospective observational study	600	Kidney transplant recipients	Only noxPTH, but not oxPTH or iPTH, was associated with graft loss in stable kidney transplant recipients	Follow-up for graft loss for 3 years
6	[49]	Prospective observational study	340	Hemodialysis patients	Measurements of noxPTH may reflect the hormone status more precisely. The iPTH-associated mortality most likely describes oxidative-stress-related mortality.	The follow-up period was 5 years
7	[41]	Retrospective studies in 2 independent cohorts	620 (4C study)	Children with CKD	In both clinical cohorts, noxPTH but not oxPTH was significantly associated with FGF23 concentrations, independent of known confounding factors.	With progressive deterioration of kidney function, total PTH and oxPTH substantially increased, whereas noxPTH only moderately increased. The increase in PTH with decreasing GFR is mainly due to an increase in oxPTH in more advanced stages of CKD.
			600	Stable renal transplant recipients		

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
