# Peer review of "Biological Activity of Different Forms of Oxidized Parathyroid Hormone"

_ijms, 2022, doi:10.3390/ijms232012228_

Round 1

Reviewer 1 Report

The authors claim that parathyroid hormone (PTH) loses its biological effects through oxidation. PTH can be oxidized at methionine 8 and 18. Three possible variations of oxidized PTH (oxPTH) exist: Met8(ox)PTH, Met18(ox)PTH, and Met8, Met18(di-ox) PTH. The currently available method which was used in these clinical studies just distinguishes between oxPTH and noxPTH without being able to differentiate between different forms of oxPTH. Only methods of PTH measurement being able to differentiate PTH forms (noxPTH, Met8(ox)PTH, Met18(ox)PTH, and Met8, Met18(di-ox) PTH) need to be developed, validated, and used in prospective randomized clinical trials to define the potential value of bioactive PTH forms as a predictor of cardiovascular events, mortality, and bone turnover. This review article explains the drawback of the current PTH test clinically. However, there are some concerns:

1.     The previous study showed a strong correlation between serum non-oxidized and total PTH, and comparable associations with histomorphometric and circulating bone turnover markers. The measuring of non-oxidized PTH using the currently available method provides no added value compared to total PTH as an indicator of bone turnover in patients with kidney failure. However, the authors state that the differentiated PTH forms (noxPTH, Met8(ox)PTH, Met18(ox)PTH, and Met8, Met18(di-ox) PTH) need to be developed, validated, and used in prospective randomized clinical trials to define the potential value of bioactive PTH forms. Further explanation is needed.

2.     Clinically, it is necessary to measure not only PTH 1-84 but also PTH fragments that are present in circulation to obtain a true biologic representation of total PTH bioactivity. How about the role of oxPTH fragments and non-oxPTH fragments?

3.     PTH 1–84 is metabolized into various PTH fragments, which are measured with varying levels of efficiency by PTH immunoassays. These PTH fragments, which increase in serum as CKD progresses contribute to CKD-associated bone disorders. High-resolution mass spectrometry for the measurement of PTH and PTH Fragments. How about the role of ox-fragment PTH level?

4.     There is limited evidence of the durable effect of parathyroidectomy (PTX) on the oxidized PTH (oxPTH) variations in dialysis populations. The authors may describe it as possible.

5.     What’s the role of oxPTH in pediatric dialysis patients?

6.     How about the path to the standardization of PTH in both CKD and non-CKD patients? Is there any difference?

Reviewer 2 Report

Paper written in a good maner and might be accepted for publication.
